# To Correct or Not Correct? Actual Evidence, Controversy and the Questions That Remain Open

**DOI:** 10.3390/jcm9061975

**Published:** 2020-06-24

**Authors:** Miguel García García, Katharina Breher, Arne Ohlendorf, Siegfried Wahl

**Affiliations:** 1Carl Zeiss Vision International GmbH, ZEISS Group, Turnstrasse 27, 73430 Aalen, Germany; arne.ohlendorf@medizin.uni-tuebingen.de (A.O.); siegfried.wahl@uni-tuebingen.de (S.W.); 2Ophthalmic Research Institute, Elfriede-Aulhorn-Straße 7, 72076 Tuebingen, Germany; katharina.breher@uni-tuebingen.de

**Keywords:** myopia, under-correction, spectacles, eye growth, short-sightedness, near-sightedness, vision

## Abstract

Clinical studies and basic research have attempted to establish a relationship between myopia progression and single vision spectacle wear, albeit with unclear results. Single vision spectacle lenses are continuously used as the control group in myopia control trials. Hence, it is a matter of high relevance to investigate further whether they yield any shift on the refractive state, which could have been masked by being used as a control. In this review, eye development in relation to eyes fully corrected versus those under-corrected is discussed, and new guidelines are provided for the analysis of structural eye changes due to optical treatments. These guidelines are tested and optimised, while ethical implications are revisited. This newly described methodology can be translated to larger clinical trials, finally exerting the real effect of full correction via single vision spectacle lens wear on eye growth and myopia progression.

## 1. Introduction

Single vision spectacle lenses are a non-invasive, simple and affordable technique for optically correcting refractive errors, such as myopia. Spectacles are well-established [1,2], and their working principle barely varies over time. Nonetheless, the WHO estimated in 2008 that around 153 million people over five years of age were considered as visually impaired as a result of uncorrected refractive errors [3].

Since the beginning of the 20th century [4,5], it has been speculated that the usage of spectacles may provoke the eyes to become lazier, weakening eyesight and subsequently inducing a higher degree of myopia. While the general public retain this idea [6], research on this topic has shown controversial results since then. During the last decades, the complex interactions of myopia and its progression have been gradually unveiled. Although some major questions still need to be solved, one main hypothesis prevails: the emmetropization process is over-ruled by a closed retinal feedback loop between blur and eye growth, mainly driven by the amplitude and sign of defocus [7]. While the exact mechanisms of how the retina interprets the blur, or whether defocus is the major contributor or just a mediator, have not been clarified yet [8], some authors have adventured into modelling in order to examine how full correction of the refractive error may disrupt this loop [9].

For instance, Medina’s model [9] proposes that continuous correction of myopia using single-vision spectacles would exacerbate the myopic condition. On the other hand, published clinical trials, which evaluated under-correction, did not find such a clear relationship, as shown in Figure 1, revealing a non-significant influence (−0.03, Z = 0.32; *p* = 0.75) on the refractive error of the eye refraction.

Still, a small trend can be elicited if the studies are sub-grouped, as shown in Appendix A. Thus, a significant change can be observed when the amount of under-correction is more than 1 dioptre ((−1 to −1.5); −0.10, Z = 2.35; *p* = 0.02), and even more when the amount exceeds 1.5 dioptres (−0.26, Z = 5.08; *p* < 0.0001), making the spectacles that are worn act in a myopigenic manner.

Furthermore, the axial length presents a non-significant change that can be observed across studies (+0.02, Z = 0.45; *p* = 0.66), as seen in Figure 2.

It is noteworthy that a few other studies that report about under-correction vs. full correction were excluded for the following reasons: (A) the possibility that monovision in monocular conditions [16] may influence eye development has not been clarified yet and, thus, it represents another confounding factor; (B) language and data availability [17] (although other studies cited it, it is only available in Japanese) and; (C) age of the subjects [18], as the emmetropization process is assumed to finish around the age of six years, with myopia onset appearing around eight years of age [19]. Therefore, an intervention before this age cannot be compared with other groups whose subjects are aged from 7 to 13 years, such as those in the included studies.

Due to such ambiguity on how spectacle wear affects eye growth, this article presents a new perspective on how to assert the short-term effects on myopia progression in an ethical, statistical and detailed form.

## 2. Study Protocol

As previously discussed by Medina 2016 [20], most studies until now have used different groups of subjects (treatment vs. control). In contrast, the authors proposed to use the same subjects before and after the subjects received their full prescription. Thus, subjects can be individuals that are newly diagnosed myopes or not fully corrected myopes who were left uncorrected for some time. The effect of spectacle lens wear on myopia progression can be then investigated by comparing the rate of myopia progression before and after the spectacles are worn. The herein presented perspective digs into this possibility and further explores the limiting commitments that a study of this type could incur.

### 2.1. Ethical Considerations

Besides following the tenets of the Declaration of Helsinki and amendments, together with the local data protection legislation; written informed consent must be obtained from the parents/guardians as well as the pupil. Therefore, accessible and understandable study descriptions and informed consents should be available, in a manner that children of all ages can fully understand the study and its consequences, being able to approve it consciously.

Nevertheless, another ethical question arises about the duration and amount of under-correction of such studies. The recent white papers from the International Myopia Institute (IMI) recommend a minimum of 3 years [19] to assert any possible efficacy on restraining myopia progression and at least 2 years more for the control of possible rebound effects. As described in this article, this represents an ethical dilemma and likewise, a compliance problem.

Firstly, it is highly unlikely that a subject that stays without being corrected for such a long period will not perceive a deterioration in their quality of life (QoL) [21]. It neither seems feasible that a participant would stop wearing the full prescription afterwards and would embrace a worsening in his/her vision during the control period. Secondly, the poor vision caused by leaving the children not fully corrected or even under-corrected should not limit, incapacitate or deteriorate school performance and daily life activities [22]. For instance, a limit could be set at 1.00 dioptre, which is similar to a deterioration of visual acuity of ≤0.5 LogMAR [23,24]. This visual acuity is at the borderline of the distance visual acuity demand for children sitting in the first row of a typically sized classroom [25].

Because of the aforementioned considerations, a study investigating the effect of full-correction needs, instead, to aim to observe the short-term effects, not only in the change on refraction but also to determine if any small cues appear on the physiological changes of the eye.

### 2.2. Subject’s Allocation and Inclusion Criteria

As a part of the study design, the inclusion criteria have to be defined. Ideally, this type of study should be multi-centric and multi-regional to avoid an after-effect because of the high regional variance of myopia prevalence, onset time and other genetic factors that can influence the results. 

The recent publications from the IMI [19] define the subjects’ inclusion criteria for myopia control studies as follows:Children should be aged from 6 to 12 years.Astigmatism should be limited to ≤−1.00 D.Anisometropia should not be greater than 1.00 D between eyes.

Nevertheless, some additional criteria should be specifically included for the current type of study:Regarding the optical correction, only myopic subjects who previously wore spectacles, but the current prescription is not matching with the actual one, or those who never wore any correction, but they require them to see clearly, might be included.Ideally, participants would be part of the recently defined ‘pre-myopia’ group or the ‘low-myopia’ groups [26].

As noted also in the publications from the IMI, subjects presenting any ocular pathology or medical conditions that affect the visual system or strabismus should be excluded.

### 2.3. Stratification—Grouping the Subjects

To limit the bias, as well as to minimise confounding factors, participants must be classified according to their refractive error. Additionally, they should be sorted by (A) whether they are already wearing spectacles, and the power does not match the actual refractive state, or (B) whether they never wore spectacle lenses before, but need them now (first-time wearers). Finally, participants needed to be grouped by age, as maturity in the contrast normalisation mechanism might not be reached until the age of 11 years old [27].

Furthermore, all the participants should be grouped by the amount of blurring they were exposed to before being corrected, as it is unclear what the exposure-response dependence is [12].

### 2.4. Randomisation, Masking and Control Group

There is no possibility of subject randomisation, as they would act as both: the control group during the period before receiving full correction via spectacles, and as the experimental group after being fully corrected.

To mask the examiners, ‘plano lenses’ might be used during the first examinations, but hardly any benefit would be obtained, as most of the measurements would be objective. On the other hand, the benefit in the visual acuity that the real correction would provide may offer enough cues to the subject to know whether he/she is corrected or not. For this reason, masking subjects in this type of study seems quite futile.

### 2.5. Scheduling and Parameters to Measure

The recent white paper from the IMI [19] summarises the main factors that should be recorded in the study protocol. For instance, they propose that the minimum dataset for studies, including spectacles, should contain at least distance and near visual acuity, pupil size measurements, cycloplegic refraction (if possible using 1% cyclopentolate [28]), optical axial length measurements, the amplitude of accommodation, contrast sensitivity tests, and wearing time assessment for compliance checks.

Whilst it is beyond doubt that all the measurements, as mentioned above, can provide insightful data, the time consumption to carry them out on a weekly or bi-weekly basis is unrealistic. Rather than gathering all measurements during every session, the following steps are proposed:

First, on every session, three measurements on each eye of full biometry, objective refraction (including aberrometry if possible) and optical coherence tomography (OCT) are performed, which altogether require no more than approximately 20 min. To avoid diurnal influences on the actual physiological changes [29,30,31], the measurements should be taken at roughly the same hour, and on a weekly or bi-weekly basis for at least three to six months, half of them uncorrected or not corrected and half of them fully corrected.

Secondly, in two sessions, one before and one after the full correction is worn, cycloplegic refraction, contrast sensitivity and accommodation measurements should be performed. The main reason why those measurements are not required on a more regular basis, but just before and after the treatment, is that no frequent temporal changes are expected, or the changes over time have been already reported [32,33]. Furthermore, the time and the distance usage of the full correction might be critical to wage myopia control efficacy, like any other effect. Ensuring that the children wear the correction at any time might be complicated; while questionnaires have been used previously to register the hours of correction wear, the reliability of such surveys is in doubt [34]. Nowadays, monitoring devices can be used for tracking the wearing time, which also provides more insights in regard to time spent performing outdoor activities or levels of light exposure [35,36].

Additional (if possible) measurements of the retinal shape and peripheral refraction might be performed before and after correction, using devices such as peripheral photorefractors [37], open-view Hartmann–Shack sensors [38], or perhaps in the future, with commercially available devices [39,40]. Reading speed test could also be included to objectively determine the QoL [41] change due to the under-correction.

### 2.6. Statistical Analysis

Quasi-experimental studies, as the design presented here, tend to use an analysis of the difference in differences (DID or DD), which compares the estimated growth based at the baseline mean and the post-effect mean. However, as a series of several temporal measurements, this approach requires a correction factor for time. Still, it is unknown which factors should be used and to what extent these factors are changing due to the natural growth of the children. The variations in the parameters (pre- to post-wear) should, therefore, be evaluated by a comparison of the slopes or trend changes of the measured variables during the first period (pre-wear) against the subsequent slopes during the next period (post-wear). Unlike the analysis of the means, this methodology would mitigate errors, as most of the variables that can be affected by spectacle wear are still in development, which leads to an offset and a skewed distribution.

Two main options can be utilised to determine the statistical differences between the slopes of pre-wear and post-wear. The first is a non-linear mixed-effects model that is also known as a hierarchical nonlinear model, which can account for individual behaviours and the highly inter-subject variance that is usually found across subjects. However, it requires extensive statistical knowledge to define covariates accurately.

Alternatively, a bi-variate correlated comparison of the slopes can be used, for which Fisher’s Z [42,43] and the Steiger/Meng’s Z statistical tests [44,45] are recommended. This approach is more simplistic than the first but can get confounded by other factors. However, it can be useful if significant changes in the slope of any parameter that show differences pointing in the same direction are found. In order to use it on a restrictive mode, not only the slopes should change, but the post-wear data should also fall outside the estimated 95% confidence interval for the normal behaviour. To calculate the confidence interval, permutation or bootstrapping techniques can be used [46].

Furthermore, when reporting the changes in all measured variables, the following is recommended: (A) using absolute rather than relative values, as this could avoid fast-progressors misleading results [47,48,49,50]; (B) including the 95% confidence intervals for any effect [19].

## 3. Proof of Concept and Feasibility

Besides the aforementioned theoretical framework drawn, a proof of concept (PoC) was carried out, in which the viability of such a study was evaluated.

### 3.1. Screening and Subjects Enrollment

Pupils from fifth to seventh grade of a local school were invited to participate in a screening program. Non-cycloplegic distance auto-refraction (NCAR) was assessed using an open-field autorefractor (WAM-5500, Grand Seiko Co. Ltd., Hiroshima, Japan) [51].

During objective autorefraction, children were briefed to look on the smallest letters of a standardised acuity chart that was placed at a distance of 4 m. After autorefraction, monocular uncorrected as well as corrected distance visual acuity (UCDVA, CDVA) were assessed using a software-based acuity test at the same distance. In cases in which autorefraction revealed a spherical equivalent refractive error (SE) < −0.5 D, monocular uncorrected distance visual acuity was <0.1 logMAR, and visual acuity increased after correction of the SE, children were classified as myopic.

In total, 472 children were screened on two different stages, the average test time being around 2 min per child. In 51 children (10.8%), the screening results revealed undiscovered and uncorrected myopia, and these children were assigned for additional examinations. Surprisingly, the prevalence of distance refractive errors was quite low for the total group of children (myopia: 16%, emmetropia 74%, hyperopia: 10%).

The prevalence of newly diagnosed myopia in this cohort of school children was around 10%, while the overall incidence was in-line with previously reported data in Germany [52]. Herein, we encounter the first possible complication, that is, the prevalence of undisclosed refractive errors. Countries like Germany, where the incidence of myopia is low and there is a high awareness of visual problems are more proactive to conduct regular ophthalmic check-ups. Therefore, the prevalence of uncorrected refractive errors (URE) will be lower, decreasing the possibility to enrol subjects.

### 3.2. Methodology

From the screened children, all 51 uncorrected myopic subjects were invited for an additional eye examination. Besides a 92% drop-off (probably because they decided to go to their local eye doctor/optician, lack of time/interest or both), 5 children agreed to be further explored. During this first visit, the longitudinal study was explained in detail and proposed to the parents and the child. Subjective refraction, ophthalmological examination as well as metrological measurements of the eye were taken during this session to confirm the findings and to show what kind of measurements would be taken during the study. 

Finally, three children and their legal guards agreed to participate in the study. These children were measured weekly with an OCT (Cirrus HD-OCT 5000, Carl ZEISS Meditec, Dublin, OH, USA), an aberrometer (iProfiler Plus, Carl ZEISS Vision Care, Aalen, Germany) and a biometer (IOL Master 700, Carl ZEISS Meditec, Dublin, OH, USA) over the course of 12 weeks, 6 before being fully corrected and 6 afterwards. Furthermore, subjective refraction and peripheral refraction using an eccentric photorefractor [37] were completed during a session before and after receiving the full-correction spectacles.

A second drawback appeared here: Although the participants were rewarded and received new spectacles without cost, the overall willingness to participate in the study was very low. Weekly measurements, as performed, seemed to be relatively time-consuming and associated with a high organisational demand in regard to coordinating their school and spare time activities. Thus, instead of weekly basis measurements, we recommend performing bi-weekly measures which could facilitate the compliance of the subjects. Subjects of this age are not likely to attend scheduled appointments on a regular and reliable basis. The rate of failure to attend on the defined time schedule was of 11%.

The screening and the pilot study were in full compliance with the Helsinki declaration (1975) and posterior amendments as well as the ethical approval to perform the measurements, and were granted by the Research Ethics Committee of the University of Tuebingen (350/2016BO1). Written informed consent was obtained from both the legal tutors/parents and the children. All data were stored and analysed in full compliance with the principles of the Data Protection Act GDPR 2016/679 of the European Union [53].

## 4. Discussion

Single vision spectacles have been established for a long time as the gold standard for correcting distance refractive errors [1], namely, myopia. Thus, it is of little surprise that have been widely used as a control means for myopia control interventional studies [54]. However, the effectiveness or countereffect in the progression of myopia is far from being known. Previous attempts to define their effect have used under-correction as a naïve direct comparison [55], but the results shed more questions than answers [20].

Furthermore, the comparison between no-correction and under-correction introduces several handicaps to the subject that have to be considered. Even in the case of slightly less myopic progression, children who are not fully corrected may face other problems [56,57,58], such as educational delays or the risk of suffering from bullying or ostracism from their peers [58]. Thus, another variable needs to be considered, weighing up the risks and ethical correctness of not correction against its benefit. Unless there is no clear inhibition/slowing down of myopia progression from under-correction, it should not be recommended as a first-choice intervention strategy, especially since there are alternatives that have proven capable of slowing the rate of myopia progression [54]. In this review, a different approach to determine the effect of full-correction with single vision spectacle wear is described, redefining how this research can be completed in the future.

The proposed methodology to assess the short-term effect of spectacle wear on the progression of myopia is not trivial, and several factors, such as the timing of measurements or the children’s growth, may confound the results. Many other factors need to be corrected, given that eye growth and day activities may vary during the scholar period compared to the school break [59]. Nonetheless, the main handicap might be that the proposed timeframe is too short for detecting refractive changes, despite the fact that it has been proven that the introduced methodology is able to acknowledge short-term changes.

Several drawbacks appeared during the pilot study, most of them being related to the subject’s compliance. Nonetheless, these led to a better understanding of how to improve the study protocol. For instance, a new timeframe is proposed with bi-weekly instead of weekly measurement appointments. The study location and place of measurement do not seem trivial for success. Therefore, it is recommended to measure in areas where enrolment is favourable and where functional cooperation networks with schools are accounted for. Moreover, the measurements should be performed within the school to achieve higher compliance rates.

## 5. Conclusions

To conclude, until more evidence is obtained, the ethical limitations surrounding under-correction or non-correction and the unclear outcomes from the literature should preclude the clinical practice from recommending leaving the subjects not fully corrected. The presented review proposes a minimum viable study protocol to deduct the effect of spectacle wear on the progression of myopia, without a broad interference in the quality of life of the subjects.

## Figures and Tables

**Figure 1 jcm-09-01975-f001:**
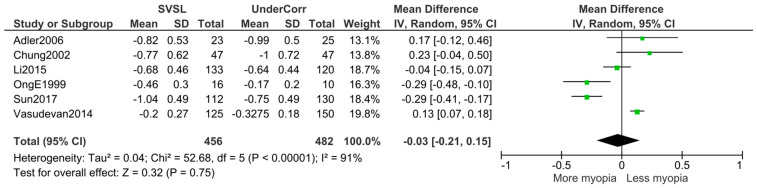
Forest plot displaying the refractive shift between different studies that reported data on under-correction versus single vision spectacle wear. Adler 2006 [10], Chung 2002 [11], Li 2015 [12], Ong 1999 [13], Sun 2017 [14] and Vasudevan 2014 [15]. Plotted using Review Manager (RevMan 5.3. Copenhagen: The Nordic Cochrane Centre, The Cochrane Collaboration).

**Figure 2 jcm-09-01975-f002:**
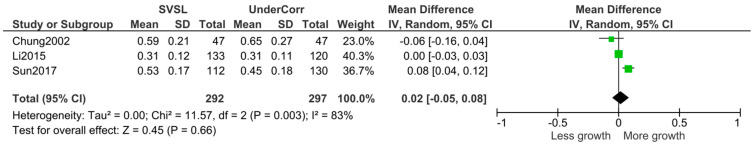
Forest plot displaying the axial length shift between different studies that reported data on under-correction versus single vision spectacle wear. Articles in the table: Chung 2002 [11], Li 2015 [12], Sun 2017 [14]. Plotted using Review Manager (RevMan 5.3. Copenhagen: The Nordic Cochrane Centre, The Cochrane Collaboration).

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
