# Peer review of "To Correct or Not Correct? Actual Evidence, Controversy and the Questions That Remain Open"

_jcm, 2020, doi:10.3390/jcm9061975_

Round 1

Reviewer 1 Report

The paper is based on a good premise, that we have not proven what the effect of wearing glasses is on the rate of progression of myopia. The conclusions in the paper are: 1) that there is no reason not to prescribe the full myopic correction to children who are myopic and 2) the study period may be too short to draw a solid conclusion. But,  I feel the main point lacking in the paper is an acknowledgement that there are other interventions (not glasses, but Atropine, OrthoK and Soft MF lenses) that have been shown to slow the rate of myopic progression more than in those wearing spectacles of full or partial correction prescription.

Introduction

In the first line, you can not say spectacles are risk-free. Although glasses provide protection from small projectile objects, there are numerous instances of children who are injured from an object hitting glasses (soccer ball, baseball, etc.) and incurring injuries due to wearing glasses. Also, the premise of the paper is in question as a risk. Does wearing glasses induce faster increases in axial length than other forms of correction? There is a potential risk in wearing glasses that accelerate the rate of axial length increase.

Minor spelling issues:

Enroll is with 2 “L’s”, also enrollment.

There are a number of minor grammatical errors that require editing by a native English speaker who is familiar with this topic.

Reviewer 2 Report

I read with interest the title of you article, but unfortunately by reading it I'm not even sure what is the purpose of this article. I was expecting an answer to question you were asking in the title, instead you are describing a kind of study protocol. However it is nicely written, I liked much some aspect of your article, for instance the ethical considerations etc., but somehow misleading. Thats why, the most important thing is to adjust the title.

Round 2

Reviewer 1 Report

I thank the authors for submitting the revised manuscript. I feel it reads better and clarifies some of the potential points of confusion.

I look forward to seeing this published soon.